# Synthesis and Investigation of Tetrahydro-β-carboline Derivatives as Inhibitors of Plant Pathogenic Fungi

**DOI:** 10.3390/molecules26010207

**Published:** 2021-01-03

**Authors:** Koonchira Buaban, Weerachai Phutdhawong, Thongchai Taechowisan, Waya S. Phutdhawong

**Affiliations:** 1Department of Chemistry, Faculty of Science, Silpakorn University, Nakhon Pathom 73000, Thailand; koonchira.buaban@gmail.com; 2Department of Science, Faculty of Liberal Arts and Science, Kamphaeng Sean Campus, Kasetsart University, Nakhon Pathom 73140, Thailand; phutdhawong@gmail.com; 3Department of Microbiology, Faculty of Science, Silpakorn University, Nakhon Pathom 73000, Thailand; tewson84@hotmail.com

**Keywords:** tetrahydro-β-carbolines (THβCs), plant pathogenic fungi, radial growth, *N*-substituted THβCs

## Abstract

A series of tetrahydro-ß-carbolines substituted with an alkyl or acyl side chain was synthesized and screened for its antifungal activity *against* plant pathogenic fungi (*Bipolaris oryzae*, *Curvularia lunata*, *Fusarium semitectum,* and *Fusarium fujikuroi*). The structure activity relationship revealed that the substituent at the piperidine nitrogen plays an important role for increasing antifungal activities. In this series, 2-octyl-2,3,4,9-tetrahydro-1*H*-pyrido[3,4-*b*]indole (**3g**) displayed potent antifungal activities with a minimum inhibitory concentration of 0.1 μg/mL, including good inhibitory activity to the radial growth of fungus at a concentration of 100 μg/mL compared to amphotericin B.

## 1. Introduction

Plant pathogenic fungi are a major cause of plant diseases and pose enormous problems for agriculture worldwide. To control and curb their spread, many natural and synthetic pesticides have been applied as preventive and therapeutic usages and are crucial for the reliable production of crops. However, a long-standing use of fungicides can lead to the plant pathogens resistance. The prime focus of the agriculture sector is to develop potent fungicides to combat plant pathogenic fungi, which are growing at an alarming pace with a limited choice of available fungicides. Hence, modification of fungicides to improve the antifungal potential is a pressing need in modern agriculture.

Naturally occurring compounds have always played an important role in drug discovery. In particular, the β-carboline based moiety represents the core structure of several pharmaceutical activities [1,2,3,4,5,6,7,8,9,10,11,12], such as anticancer [3,4,13], antimicrobial [5], antimalarial [7], antiviral [9], and antifungal activities [14,15,16,17,18,19,20]. Many bioactive β-carboline-based compounds have been found to be important sources of drugs and drug leads, and most of them have been isolated from marine invertebrates, such as the marine sponge [5,6,7,8,10,21,22,23], tunicate [4,24,25], gorgonian [26] and alga [27]. For example, as shown in Figure 1, Eudistomins and their analogs were isolated from several species of marine organisms, mainly the active Caribbean colonial tunicate, and possessed a wide range of biological activities [28,29,30,31,32]. Ingenines, β-carboline alkaloids with a saturated/unsaturated tricyclic ring system, were isolated from the Indonesian marine sponge *Acanthostrongylophora ingens* and exhibited promising candidates for potent cytotoxic agents [13,33,34]. 6-Chloro-9-(3-(4-chlorophenoxy)propyl)-2,3,4,9-tetrahydro-1*H*-pyrido [3,4-*b*]indole TFA displayed in vitro and in vivo antifungal activity of *Cryptococcus neoformans* by induction of cell growth at the G2 phase on the Cdc25c/CDK1/cyclin B pathway [18]. Moreover, THβC with an eight-carbon linear chain at the C1 position, (1*R*,3*S*)-methyl 1-octyl-2,3,4,9-tetrahydro-1*H*-pyrido[3,4-*b*]indole-3-carboxylate, showed in vitro antifungal activity *against Candida glabrata* with disruption of the membranes of fungal cells by increased asymmetry and decomposition of the cell surface [20]. Thus, the pharmaceutically important β-carboline-based natural products are advantageous for medicinal and organic research, while 1,3 disubstituted THβCs are reported to be synthetically and biologically interesting analogs [35]. Currently, Tadalafil, the hydantoin fused at the nitrogen and C3 of THβC with a substituent at C1 is clinically used to treatment erectile dysfunction under the brand name Cialis [36] and is also used for pulmonary arterial hypertension treatment under the brand name Adcirca [37]. In our on-going search for novel compounds as inhibitors of plant pathogenic fungi, we are interested in C1 substituted THβCs as the skeleton, which is a common feature in many natural and synthetic compounds. Furthermore, investigations have indicated that nitrogen substituted with a longer alkyl or acyl side chain was an essential framework in various antifungal agents [38,39,40,41,42,43].

Thus, in this study, a series of *N*-substituted THβCs with an alkyl or acyl substituent together with C1 and/or C3 substituted THβCs was designed and synthesized from tryptamine and various aldehydes via the Pictet-Spengler reaction. *N*-substitutions of THβCs were prepared to evaluate their inhibition of four plant pathogenic fungi (*Bipolaris oryzae, Curvularia lunata, Fusarium semitectum*, and *Fusarium fujikuroi*).

## 2. Results and Discussion

### 2.1. Chemistry

The incorporation of aromatic substituents at the C1 position was reported to improve the activity, and the presence of the carbonyl group at the C3 position was also found to be important for the enhancement of activity [1,11,21,27,35]. Thus, we designed the preparation of the substituted THβCs via the efficient Pictet-Spengler reaction with tryptamine (**1a**), *L*-tryptophan methyl ester (**1b**), and appropriate aldehydes in acid media (Scheme 1). The THβCs (**2a**–**2h**) were obtained in poor to moderate yields. The steric hindrance and the presence of the electron-donating group at *ortho* position on the aromatic ring reduced the electrophilicity of aldehydes [44] and affected the yield of the THβC ring, while the less hindered electron donating group induced the cyclization to the THβC skeleton. The methyl ester substituent also increased the cyclization process. Next, the N2-substituted-THβC bearing an alkyl or acyl side chain (**3a**–**3k**) was prepared from an acyl halide or alkyl halide in the presence of NEt_3_ as a base. *N*-alkylated and *N*-acylated products were achieved in moderated to high yields with the exception of 2-furoyl substituent (**3e**). The brown oil of compound (**3e**) was obtained in low yield due to the difficulty of purification. All synthesized compounds were characterized via MS, ^1^H-NMR, and ^13^C-NMR. All obtained THβCs were the racemic mixture at C1 and were used for antifungal evaluation.

### 2.2. Antifungal Activity

All THβC derivatives were evaluated for their antifungal activity *against* four phytopathogenic fungi of rice, *Bipolaris oryzae*, *Curvularia lunata*, *Fusarium semitectum*, and *Fusarium fujikuroi*, using the results presented in Table 1. Amphotericin B and solvent were used as a positive and negative control, respectively. Most of the tested compounds showed activities for inhibition of *B. oryzae* and *C. Lunata*, while most of *N*-substitued THβCs were active against all tested fungi. The aromatic substituents at C1 of THßCs (**2b**–**2f**) showed a moderate activity *against B. oryzae* and *C. lunata* (zone of inhibition = 0.10–0.60 cm), whereas C1, C3-disubstituted THßCs (**2h**) showed a slight increase in inhibitory activity compared to that of C1-substituted THßC (**2b**). The substituent at N2 of THßCs (**3a**–**3k**) was investigated. The linear acyl chains varying in length from 2 to 8 carbons (C2–C8), compounds (**3a**–**3c**), possessed no inhibitory activity. In contrast, the aromatic acyl groups (**3d**–**3e**) and alkyl chain substituents (**3f**–**3i**) led to increased activity with a zone of inhibition value of 0.10–0.60 cm. Thus, this preliminary in vitro antifungal activity indicated that substituent at N2 position with aromatic acyl groups and alkyl chains was found to be helpful for the board spectrum antifungal activity. Therefore, the minimum inhibitory concentration (MIC) was determined for *N*-substituted THßCs (**3a**–**3k**). The MIC values of compounds (**3d**–**3i**) were 28–520 μg/mL against *B. oryzae* and *C. lunata*. Interestingly, the *N*-octyl derivative (**3g**) showed the highest activity against *B. oryzae* (MIC = 28 μg/mL), whereas the C1-phenyl and *N*-octyl derivative (**3k**) was less active than (**3g**) *against B. oryzae, C. lunata,* and *F. semitectum*. It can be indicated that racemic mixture of phenyl substituent at C1 was not favorable for the fungicidal activity and *N*-substituted THßCs play an essential role in antifungal activity. However, further studies of the enantiopure form of the C1 and *N*-substituted THβCs were under investigation.

Inhibitory activity of the *N*-octyl derivative (**3g**) was studied *against* a diameter of a colony of phytopathogenic fungi at concentration ranges of 0, 100, 250, 500, and 1000 μg/mL. The results in Table 2 and Figure 2 show that compound (**3g**) at concentration of 100 µg/mL inhibited the fungal growth by 68.2%, 37.4%, and 40% of *B. oryzae*, *F. Fujukuroi*, and *C. lutana,* respectively. However, at this concentraton, compound (**3g**) could not inhibit the growth of *F. Semitectum*. Although, *N*-octyl substituted THβC displayed less active than the antifungal drug Amphotericin B. The chemical modification of *N*-alkyl THßCs will provide the novel chemical scaffolds for further structural optimization of phytopathogenic fungi in the future.

## 3. Materials and Methods

### 3.1. Chemistry

#### 3.1.1. General Information

The melting points were determined on an SMP2 model melting point apparatus and were uncorrected. The ^1^H and ^13^C spectra were recorded in a Bruker AVANCE 300 MHz nuclear magnetic resonance spectrometer with CDCl_3_, CD_3_OD, and (CD_3_)_2_SO as the solvent and TMS as the internal standard. The chemical shifts were presented in parts per million (ppm, δ), the coupling constants (*J*) were reported in hertz (Hz), and the signals were described as singlet (s), doublet (d), triplet (t), quartet (q), quintet (qui), broad singlet (br s), doublet of doublet (dd), doublet of triplet (dt), triplet of doublet (td), doublet of doublet of doublet (ddd), doublet of doublet of triplet (ddt), triplet of doublet of doublet (tdd), and multiplet (m). HR-MS was performed using Bruker Daltonics. All compounds were monitored using a TLC silica gel 60 F254 aluminum sheet. Column chromatography was performed using silica gel 60 (0.063–0.200 mm) and visualized under UV light at 254 and 365 nm. All chemicals were purchased from commercial suppliers.

#### 3.1.2. General Procedure for the Synthesis of 1-substituted-tetrahydro-β-carbolines (**2a**–**2h**)

Tryptamine (**1a**) or tryptophan methyl ester (**1b**) was dissolved in AcOH: dry CH_2_Cl_2_ (5:10 mL) in a round-bottom flask. Then, aldehyde (1.2 eq) was slowly added to the solution, and the solution was refluxed for 1–2 h. After completion, the reaction mixture was cooled to room temperature and basified to pH 9–10 using NH_4_OH. Afterwards, the solution was extracted with CH_2_Cl_2_, and the organic layer was dried over anh. Na_2_SO_4_ and evaporated under reduced pressure. The crude product was purified using column chromatography (silica gel, 15% MeOH:CH_2_Cl_2_).

2,3,4,9-Tetrahydro-1*H*-pyrido[3,4-*b*]indole (**2a**). The title compound was synthesized from (**1a**) (268 mg, 1.7 mmol) and *para*-formaldehyde (60 mg, 2.0 mmol) to afford (**2a**) (153 mg, 53%) as a yellow solid. m.p. 185.9–187.5 °C (lit. 109–221 °C) [15,45]; ^1^H-NMR (300 MHz, CDCl_3_) δ 2.75 (t, *J* = 5.7 Hz, 2H), 3.18 (t, *J* = 5.7 Hz, 2H), 4.01 (s, 2H), 7.09 (dt, *J* = 1.1, 7.1 Hz, 1H), 7.14 (dt, *J* = 1.3, 7.1 Hz, 1H), 7.29 (d, *J* = 7.2 Hz, 1H), 7.48 (d, *J* = 7.2 Hz, 1H), 7.86 (br s, 1H); ^13^C-NMR (125 MHz, CDCl_3_) [46] δ 22.5 (CH_2_), 43.2 (CH_2_), 43.9 (CH_2_), 108.7 (C), 110.7 (CH), 117.9 (CH), 119.4 (CH), 121.5 (CH), 127.6 (C), 132.7 (C), 135.6 (C).

1-Phenyl-2,3,4,9-tetrahydro-1*H*-pyrido[3,4-*b*]indole (**2b**). The title compound was synthesized from (**1a**) (321 mg, 2.0 mmol) and benzaldehyde (0.24 mL, 2.4 mmol) to afford (**2b**) (124 mg, 25%) as a yellow solid. m.p. 162.9–164.7 °C (lit. 160–161 °C) [47]; ^1^H-NMR (300 MHz, CDCl_3_) δ 1.97 (br s, 1H), 2.78–3.00 (m, 1H), 3.07–3.18 (m, 1H), 3.37 (dt, *J* = 3.9, 12.5 Hz, 1H), 5.17 (s, 1H), 7.10 (dt, *J* = 1.9, 6.9 Hz, 1H), 7.14 (dt, *J* = 1.9, 6.9 Hz, 1H), 7.17–7.23 (m, 1H), 7.27–7.37 (m, 5H), 7.52–7.56 (m, 1H); ^13^C-NMR (75 MHz, CDCl_3_) δ 22.5 (CH_2_), 42.9 (CH_2_), 58.1 (CH), 110.2 (C), 110.8 (CH), 118.3 (CH), 119.4 (CH), 121.8(CH), 127.4 (C), 128.2 (CH), 128.5 (2CH), 128.8 (2CH), 134.4 (C), 135.9 (C), 141.7 (C).

1-(2,5-Dimethoxyphenyl)-2,3,4,9-tetrahydro-1*H*-pyrido[3,4-*b*]indole (**2c**). The title compound was synthesized from (**1a**) (494 mg, 3.1 mmol) and 2,5-dimethoxybenzaldehyde (615 mg, 3.7 mmol) to afford (**2c**) (86 mg, 9% yield) as a yellow solid. m.p. 132.2–133.6 °C; ^1^H-NMR (300 MHz, CDCl_3_) δ 2.87–2.92 (m, 2H), 3.15–3.25 (m, 1H), 3.29–3.39 (m, 1H), 3.67 (s, 3H), 3.84 (s, 3H), 5.65 (s, 1H), 6.68 (d, *J* = 2.9 Hz, 1H), 6.80–6.93 (m, 2H), 7.10 (dt, *J* = 1.4, 7.1 Hz, 1H), 7.14 (dt, *J* = 1.4, 7.0 Hz, 1H), 7.22–7.24 (m, 1H), 7.52 (d, *J* = 7.0 Hz, 1H), 7.83 (br s, NH); ^13^C-NMR (75 MHz, CDCl_3_) δ 21.1 (CH_2_), 21.4 (CH_2_), 41.0 (CH_2_), 51.3 (CH), 55.8 (CH_3_), 56.1 (CH_3_), 109.6 (C), 110.9 (CH), 111.9 (CH), 113.9 (CH), 116.1 (CH), 118.3 (CH), 119.5 (CH), 121.9 (CH), 127.0 (C), 128.5 (C), 136.0 (C), 151.5 (C), 153.7(C), 175.9 (C); HREI-MS: calcd for C_19_H_20_N_2_O_2_ [M]^+^: 308.1525, found: 308.1527.

1-(3-Methoxyphenyl)-2,3,4,9-tetrahydro-1*H*-pyrido[3,4-*b*]indole (**2d**). The title compound was synthesized from (**1a**) (303 mg, 1.9 mmol) and 3-methoxybenzaldehyde (0.20 mL, 2.3 mmol) to afford (**2d**) (385 mg, 73% yield) as an orange solid. m.p. 150.8–153.8 °C (lit. 154–156 °C) [48]; ^1^H-NMR (300 MHz, CDCl_3_) δ 2.82–3.04 (m, 2H), 3.10–3.22 (m, 1H), 3.32–3.43 (m, 1H), 3.75 (s, 3H), 5.32 (s, 1H), 6.84–6.91 (d, *J* = 3.9 Hz, 3H), 7.11 (dt, *J* = 1.5, 7.0 Hz, 1H), 7.16 (dt, *J* = 1.5, 7.0 Hz, 1H), 7.22 (d, *J* = 2.0 Hz, 1H), 7.23–7.29 (m, 1H), 7.53 (dd, *J* = 2.0, 7.0 Hz, 1H), 7.80 (br s, NH); ^13^C-NMR (75 MHz, CDCl_3_) δ 21.3 (CH_2_), 41.7 (CH_2_), 55.3 (CH_3_), 57.2 (CH), 109.8 (C), 111.0 (CH), 111.5 (CH), 114.2 (CH), 118.3 (CH), 119.6 (CH), 121.0 (CH), 122.1 (CH), 127.0 (C), 129.9 (CH), 132.3 (C), 136.1 (C), 140.9 (C), 160.1 (C); HRESI-MS: calcd for C_18_H_18_N_2O_ [M+H]^+^: 279.1492, found: 279.1486.

1-(2-Methoxyphenyl)-2,3,4,9-tetrahydro-1*H*-pyrido[3,4-*b*]indole (**2e**). The title compound was synthesized from (**1a**) (141 mg, 0.9 mmol) and 2-methoxybenzaldehyde (144 mg, 1.1 mmol) to afford (**2e**) (22 mg, 9% yield) as an orange solid. m.p. 97.8–99.3 °C (lit. 95–96 °C) [49]; ^1^H-NMR (300 MHz, CDCl_3_) δ 2.28 (br s, 1H), 2.81 (d, *J* = 4.9 Hz, 2H), 3.00–3.11 (m, 1H), 3.16–3.28 (m, 1H), 3.85 (s, 3H), 5.57 (s, 1H), 6.83 (t, *J* = 7.4 Hz, 1H), 6.92 (d, *J* = 8.2 Hz, 1H), 6.98 (dd, *J* = 1.4, 7.5 Hz, 1H), 7.06–7.18 (m, 3H), 7.26 (dd, *J =* 1.5, 15.6 Hz, 1H), 7.50 (d, *J =* 3.7 Hz, 1H), 7.94 (s, NH); ^13^C-NMR (75 MHz, CDCl_3_) δ 22.4 (CH_2_), 41.7 (CH_2_), 50.9 (CH_3_), 55.5 (CH), 110.2 (C), 110.7 (CH), 110.8 (CH), 118.0 (CH), 119.2 (CH), 120.6 (CH), 121.4 (CH), 127.3 (C), 129.0 (CH), 129.2 (CH), 129.7 (C), 134.1 (C), 135.8 (C), 157.2 (C).

1-Phenethyl-2,3,4,9-tetrahydro-1*H*-pyrido[3,4-*b*]indole (**2f**). The title compound was synthesized from (**1a**) (489 mg, 3.1 mmol) and 3-phenylpropanal (491 mg, 3.7 mmol) to afford (**2f**) (396 mg, 47% yield) as an orange oil. ^1^H-NMR (300 MHz, CDCl_3_) δ 1.89–2.04 (m, 1H), 2.05–2.20 (m, 1H), 2.68–2.76 (m, 2H), 2.76–2.92 (m, 2H), 2.96–3.08 (m, 1H), 3.33 (td, *J* = 4.7, 12.8 Hz, 1H), 4.06 (d, *J* = 5.2 Hz, 1H), 7.07 (dt, *J* = 1.3, 7.1 Hz, 1H), 7.12 (dt, *J* = 1.3, 7.0 Hz, 1H), 7.16–7.33 (m, 6H), 7.47 (d, *J* = 7.0 Hz, 1H), 7.74 (br s, NH); ^13^C-NMR (75 MHz, CDCl_3_) δ 22.7 (CH_2_), 29.7 (CH_2_), 36.7 (CH_2_), 42.4 (CH_2_), 52.2 (CH), 109.1 (C), 110.7 (CH), 118.0 (CH), 119.0 (CH), 121.5 (CH), 126.0 (CH), 127.5 (C), 128.4 (2CH), 128.5 (2CH), 135.6 (C), 135.9 (C), 141.9 (C). HRESI-MS: calcd for C_19_H_20_N_2_ [M+H]^+^: 277.1700, found: 277.1706.

Methyl 2,3,4,9-tetrahydro-1*H*-pyrido[3,4-*b*]indole-3-carboxylate (**2g**). The title compound was synthesized from (**1b**) (271 mg, 1.2 mmol) and *para*-formaldehyde (45 mg, 1.5 mmol) to afford (**2g**) (197 mg, 69% yield) as a yellow solid. m.p. 171.8–173.6 °C (lit. 187.2–188.8 °C) [50]; ^1^H-NMR (300 MHz, CDCl_3_) δ 2.05 (br s, NH), 2.89 (tdd, *J =* 1.5, 4.2, 15.2 Hz, 1H), 3.14 (tdd, *J* = 1.5, 4.2, 15.2 Hz, 1H), 3.79 (s, 3H), 3.81–3.85 (m, 1H), 4.08–4.30 (m, 2H), 7.10 (dt, *J* = 1.3, 7.0 Hz, 1H), 7.15 (dt, *J* = 1.3, 7.0 Hz, 1H), 7.30 (d, *J* = 8.0 Hz, 1H), 7.48 (d, *J =* 7.4 Hz, 1H), 7.87 (br s, NH); ^13^C-NMR (75 MHz, CDCl_3_) δ 24.2 (CH_2_), 40.7 (CH_2_), 51.1 (CH_3_), 54.7 (CH), 105.5 (C), 109.8 (CH), 116.6 (CH), 118.2 (CH), 120.5 (CH), 126.0 (C), 130.6 (C), 135.0 (C), 172.7 (C).

Mixture of methyl 1-phenyl-2,3,4,9-tetrahydro-1*H*-pyrido[3,4-*b*]indole-3-carboxylate (**2h**). The title compound was synthesized from (**1b**) (199 mg, 9.1 mmol) and benzaldehyde (0.11 mL, 1.1 mmol) to afford (**2h**) (224 mg, 80% yield) as a brown oil. ^1^H-NMR (300 MHz, CDCl_3_) δ (*cis* isomer) 3.11 (ddt, *J* = 1.2, 6.9, 8.1 Hz, 1H), 3.26 (ddt, *J* = 1.0, 5.3, 8.0 Hz, 1H), 3.69 (s, 3H), 3.94 (t, *J* = 6.6 Hz, 1H), 5.23 (s, 1H), 7.04–7.10 (m, 2H), 7.11–7.19 (m, 1H), 7.20–7.27 (m, 4H), 7.49–7.94 (m, 1H), 7.65–7.76 (m, 1H); ^13^C-NMR (75 MHz, CDCl_3_) δ (*cis* isomer) 25.7 (CH_2_), 52.1 (CH_3_), 56.8 (CH), 58.6 (CH), 108.7 (C), 110.9 (CH), 118.1 (CH), 119.4 (CH), 121.8 (CH), 127.0 (C), 128.0 (CH), 128.4 (CH), 128.6 (2CH), 128.7 (CH), 134.6 (C), 136.1 (C), 140.7 (C), 173.2 (C); ^1^H-NMR (300 MHz, CDCl_3_) δ (*trans* isomer) 3.01 (ddd, *J* = 2.6, 7.7, 15.1 Hz, 1H), 3.23 (ddd, *J* = 1.9, 4.3, 15.1 Hz, 1H), 3.69 (s, 3H), 3.94 (t, *J* = 6.6 Hz, 1H), 5.11 (s, 1H), 7.04–7.10 (m, 2H), 7.11–7.19 (m, 1H), 7.27–7.45 (m, 4H), 7.49–7.94 (m, 1H), 7.82–7.97 (m, 1H); ^13^C-NMR (75 MHz, CDCl_3_) δ (*trans* isomer) 24.7 (CH_2_), 52.1 (CH_3_), 52.3 (CH), 54.8 (CH), 108.3 (C), 110.9 (CH), 118.1 (CH), 119.5 (CH), 121.8 (CH), 126.9 (C), 128.0 (CH), 128.4 (CH), 128.6 (2CH), 128.7 (CH), 133.1 (C), 136.1 (C), 141.9 (C), 174.0 (C).

#### 3.1.3. General Procedure for Preparation of 2-substituted-2,3,4,9-tetrahydro-1H-pyrido[3,4-b]indole (**3a**–**3i**)

2,3,4,9-Tetrahydro-1*H*-pyrido[3,4-*b*]indole (**2a**) (1 eq) was dissolved in CH_2_Cl_2_ (1 mL/mmol), and then NEt_3_ (3 eq) was added to the solution. An acid halide or acyl halide was slowly added to the solution mixture. After stirring at room temperature for 24 h, the product was extracted with CH_2_Cl_2_, and the organic layer was dried over anh. Na_2_SO_4_ and evaporated under reduced pressure. The crude product was purified using column chromatography (silica gel, 5% MeOH:CH_2_Cl_2_).

2-Acetyl-1,2,3,4-tetrahydro-ß-carboline (**3a**). The title compound was synthesized from (**2a**) (150 mg, 0.90 mmol) and acetyl bromide (0.13 mL, 1.8 mmol) to afford (3a) (79 mg, 42% yield) as a yellow solid. m.p. 217.2–218.3 °C (lit. 237–238 °C) [51]; 1H-NMR (300 MHz, MeOD) δ rotamers 1/3 (from the duplicated triplet signal (1H) at 3.83 and 3.92 ppm); 1H-NMR (300 MHz, MeOD) δ (major rotamer) 2.23 (s, 3H), 2.84 (t, *J* = 5.7 Hz, 2H), 3.83 (t, *J* = 5.7 Hz, 2H), 4.75 (s, 2H), 6.98 (t, *J* = 7.6 Hz, 1H), 7.02–7.10 (m, 1H), 7.29 (dd, *J* = 4.0, 7.9 Hz, 1H), 7.40 (d, *J* = 7.6 Hz, 1H); δ (distinct peaks for minor rotamer) 2.20 (s, 3H), 2.76 (t, *J* = 5.7 Hz, 2H), 3.92 (t, *J* = 5.7 Hz, 2H), 4.72 (s, 2H); 13C-NMR (75 MHz, MeOD) δ (major rotamers) 21.4 (CH_3_), 22.7 (CH_2_), 41.4 (CH_2_), 46.1 (CH_2_), 108.2 (C), 111.9 (CH), 118.5 (CH), 119.8 (CH), 122.2 (CH), 128.2 (C), 131.4 (C), 138.0 (C), 172.4 (C); δ (distinct peaks for minor rotamer) 21.9 (CH_3_), 21.9 (CH_2_), 45.4 (CH_2_), 109.1 (C), 118.6 (CH), 119.9 (CH), 122.4 (CH), 130.8 (CH), 172.5 (C).

1-(3,4-Dihydro-1*H*-pyrido[3,4-*b*]indol-2(9*H*)-yl)propan-1-one (**3b**). The title compound was synthesized from (**2a**) (150 mg, 0.9 mmol) and propionyl chloride (0.05 mL, 0.5 mmol) to afford (**3b**) (155 mg, 78% yield) as a brown solid. m.p. 203.7–204.1 °C (lit. 204–206 °C) [52]; rotamers 1/4 (from the duplicated triplet signal (^1^H) at 3.79 and 3.97 ppm); ^1^H-NMR (300 MHz, CDCl_3_) (major rotamer) δ 1.24 (t, *J* = 7.5 Hz, 3H), 2.51 (q, *J* = 7.5 Hz, 2H), 2.86 (t, *J* = 5.6 Hz, 2H), 3.79 (t, *J* = 5.6 Hz, 2H), 4.82 (s, 2H), 7.09 (dt, *J* = 0.9, 7.6 Hz, 1H), 7.15 (dt, *J* = 1.5, 7.6 Hz, 1H), 7.33 (d, *J* = 7.7 Hz, 1H), 7.46 (d, *J* = 7.5 Hz, 1H), 8.35 (br s, 0.6H); δ (distinct peaks for minor rotamer) 2.43 (q, *J* = 7.5 Hz, 2H), 2.81 (t, *J* = 5.6 Hz, 2H), 3.97 (t, *J =* 5.6 Hz, 2H), 4.66 (s, 2H), 7.95 (br s, 0.4H); ^13^C-NMR (75 MHz, CDCl_3_) (major rotamer) δ 9.6 (CH_3_), 22.0 (CH_2_), 26.8 (CH_2_), 40.5 (CH_2_), 43.8 (CH_2_), 107.8 (C), 111.0 (CH), 117.8 (CH), 119.5 (CH), 121.7 (CH), 126.8 (C), 130.6 (C), 136.2 (C), 173.2 (C); δ (distinct peaks for minor rotamer) 40.1 (CH_2_), 43.0 (CH_2_); HRESI-MS: calcd for C_14_H_16_N_2_O [M+H]^+^: 229.1335, found: 229.1335.

1-(3,4-Dihydro-1*H*-pyrido[3,4-*b*]indol-2(9*H*)-yl)octan-1-one (**3c**). The title compound was synthesized from (**2a**) (108 mg, 0.6 mmol) and octanoyl chloride (0.21 mL, 1.2 mmol) to afford (**3c**) (125 mg, 41% yield) as a yellow solid; m.p. 149.9–150.0 °C; rotamers 1/3 (from the duplicated triplet signal (^1^H) at 3.80 and 3.97 ppm); ^1^H-NMR (300 MHz, CDCl_3_) (major rotamer) δ 0.87 (t, *J* = 1.5 Hz, 3H), 1.20–1.50 (m, 10H), 2.48 (t, *J* = 7.5 Hz, 2H), 2.86 (t, *J* = 5.4 Hz, 2H), 3.80 (t, *J* = 5.7 Hz, 2H), 4.82 (s, 2H), 7.09 (t, *J* = 7.1 Hz, 1H), 7.16 (t, *J* = 7.1 Hz, 1H), 7.33 (d, *J* = 7.1 Hz, 1H), 7.46 (d, *J* = 7.1 Hz, 1H), 8.40 (br s, NH); δ (distinct peaks for minor rotamer) 2.41 (t, *J* = 7.5 Hz, 2H), 2.82 (t, *J* = 5.4 Hz, 2H), 3.97 (t, *J =* 5.7 Hz, 2H), 4.67 (s, 2H), 7.94 (br s, NH); ^13^C-NMR (75 MHz, CDCl_3_) (major rotamer) δ 14.1 (CH_3_), 22.1 (CH_2_), 25.5 (CH_2_), 29.2 (CH_2_), 29.5 (2CH_2_), 31.7 (CH_2_), 33.7 (CH_2_), 40.5 (CH_2_), 44.0 (CH_2_), 111.0 (CH), 117.8 (CH), 119.5 (CH), 121.7 (CH), 127.0 (C), 129.0 (C), 130.7 (C), 136.2 (C), 172.5 (C); δ (distinct peaks for minor rotamer) 22.6 (CH_2_), 43.6 (CH_2_), 118.3 (CH), 119.8 (CH), 122.1 (CH); HRESI-MS: calcd for C_19_H_26_N_2_O [M+H]^+^: 299.2118, found: 299.2116.

(3,4-Dihydro-1*H*-pyrido[3,4-b]indol-2(9*H*)-yl)(phenyl)methanone (**3d**). The title compound was synthesized from (**2a**) (121 mg, 0.7 mmol) and benzoyl chloride (0.16 mL, 1.4 mmol) to afford (**3d**) (93 mg, 61% yield) as a yellow oil; rotamers 1/4 (from the duplicated singlet signal (^1^H) at 4.46 and 4.89 ppm); ^1^H-NMR (300 MHz, CDCl_3_) (major rotamer) δ 2.88 (s, 2H), 3.66 (s, 2H), 4.89 (s, 2H), 7.00–7.13 (m, 2H), 7.18 (d, *J =* 5.8 Hz, 1H), 7.31–7.61 (s, 6H), 8.95 (br s, NH); δ (distinct peaks for minor rotamer) 4.07 (s, 2H), 4.46 (s, 2H), 8.34 (br s, NH); ^13^C-NMR (75 MHz, CDCl_3_) (major rotamer) δ 22.1 (CH_2_), 41.1 (CH_2_), 46.1 (CH_2_), 107.8 (C), 111.1 (CH), 117.8 (CH), 119.6 (CH), 121.8 (CH), 126.8 (C), 126.9 (2CH), 128.6 (2CH), 130.0 (C), 130.0 (CH), 136.0 (C), 136.3 (C), 171.6 (C); HR-ESI-MS [53] C_18_H_16_N_2_O [M]: 276.1263.

(3,4-Dihydro-1*H*-pyrido[3,4-*b*]indol-2(9*H*)-yl)(furan-2-yl)methanone (**3e**). The title compound was synthesized from (**2a**) (93 mg, 0.5 mmol) and 2-furoyl chloride (0.11 mL, 1.1 mmol) to afford (**3e**) (18.7 mg, 13% yield) as a brown oil; ^1^H-NMR (300 MHz, CDCl_3_) δ 2.95 (s, 2H), 4.09 (br s, 2H), 4.92 (s, 2H), 6.51 (dd, *J* = 1.7, 3.4 Hz, 1H), 7.07 (d, *J* = 3.5 Hz, 1H), 7.09 (dt, *J* = 1.3, 7.2 Hz, 1H), 7.15 (dt, *J* = 1.3, 7.4 Hz, 1H), 7.31 (d, *J* = 7.2 Hz, 1H), 7.48 (d, *J* = 7.2 Hz, 1H), 7.54 (s, 1H), 8.40 (br, NH); ^13^C-NMR (75 MHz, CDCl_3_) δ 20.4 (CH_2_), 40.2 (CH_2_), 43.5 (CH_2_), 109.2(CH), 109.6 (CH), 112.7 (C), 114.7 (CH), 116.0 (CH), 117.9 (CH), 120.1 (CH), 125.7 (C), 128.1 (C), 135.8 (C), 142.3 (CH), 146.1 (C), 158.4 (C); HRESI-MS: calcd for C_16_H_14_N_2_O_2_ [M+H]^+^: 267.1128, found: 267.1128.

2-Pentyl-2,3,4,9-tetrahydro-1*H*-pyrido[3,4-*b*]indole (**3f**). The title compound was synthesized from (**2a**) (115 mg, 0.7 mmol) and 1-bromopentane (0.25 mL, 2.0 mmol) to afford (**3f**) (93 mg, 57% yield) as a yellow solid. m.p. 110.9–112.6 °C (lit. 119–120 °C) [54]; ^1^H-NMR (300 MHz, CDCl_3_) δ 0.92 (t, *J* = 6.8 Hz, 3H), 1.30–1.50 (m, 4H), 1.55–1.67 (m, 2H), 2.56–2.63 (m, 2H), 2.75–2.92 (m, 4H), 3.69 (s, 2H), 7.07 (dt, *J* = 1.4, 7.0 Hz, 1H), 7.12 (dt, *J =* 1.4, 7.0 Hz, 1H), 7.26–7.33 (m, 1H), 7.46 (d, *J* = 7.2 Hz, 1H), 7.69 (br s, NH); ^13^C-NMR (75 MHz, CDCl_3_) δ 12.1 (CH_3_), 19.4 (CH_2_), 20.7 (CH_2_), 25.3 (CH_2_), 27.8 (CH_2_), 48.6 (CH_2_), 49.2 (CH_2_), 56.0 (CH_2_), 106.7 (C), 108.6 (CH), 116.0 (CH), 117.3 (CH), 119.3 (CH), 130.0 (C), 134.1 (C).

2-Octyl-2,3,4,9-tetrahydro-1*H*-pyrido[3,4-*b*]indole (**3g**). The title compound was synthesized from (**2a**) (171 mg, 1.0 mmol) and 1-bromooctane (0.34 mL, 3.0 mmol) to afford (**3g**) (119 mg, 42% yield) as a yellow solid; m.p. 105.3–106.7 °C; ^1^H-NMR (300 MHz, CDCl_3_) δ 0.89 (t, *J* = 6.6 Hz, 3H), 1.23–1.37 (m, 10H), 1.61, (qui, *J* = 7.5 Hz, 2H), 2.50–2.60 (m, 2H), 2.79–2.90 (m, 4H), 3.67 (s, 2H), 7.06 (t, *J* = 7.1 Hz, 1H), 7.13 (t, *J =* 7.1 Hz, 1H), 7.28 (dd, *J* = 1.4, 6.5 Hz, 1H), 7.46 (dd, *J* = 1.4, 6.5 Hz, 1H), 7.79 (br s, NH); ^13^C-NMR (75 MHz, CDCl_3_) 14.1 (CH_3_), 21.3 (CH_2_), 22.7 (CH_2_), 27.5 (CH_2_), 27.6 (CH_2_), 29.3 (CH_2_), 29.6 (CH_2_), 31.9 (CH_2_), 50.5 (CH_2_), 51.1 (CH_2_), 57.9 (CH_2_), 108.6 (C), 110.6 (CH), 117.9 (CH), 119.3 (CH), 121.3 (CH), 127.3 (CH), 131.9 (C), 136.1 (C); HRESI-MS: calcd for C_19_H_28_N_2_ [M+H]^+^: 285.2325, found: 285.2320.

(*E*)-2-(But-2-en-1-yl)-2,3,4,9-tetrahydro-1*H*-pyrido[3,4-*b*]indole (**3h**). The title compound was synthesized from (**2a**) (310 mg, 1.8 mmol) and crotyl chloride (0.26 mL, 2.7 mmol) to afford (**3h**) (98 mg, 24% yield) as an orange solid; m.p. 82.8–84.2 °C; ^1^H-NMR (300 MHz, CDCl_3_) δ 1.72 (d, *J* = 5.0 Hz, 3H), 2.83–2.90 (m, 4H), 3.15 (d, *J* = 5.0 Hz, 2H), 3.59 (s, 2H), 5.53–5.74 (m, 2H), 7.06 (dt, *J =* 1.4, 7.0 Hz, 1H), 7.11 (dt, *J* = 1.4, 7.0 Hz, 1H), 7.24 (d, *J* = 6.8 Hz, 1H), 7.45 (d, *J* = 6.8 Hz, 1H), 8.02 (br s, NH); ^13^C-NMR (75 MHz, CDCl_3_) δ 17.8 (CH_3_), 21.1 (CH_2_), 49.9 (CH_2_), 50.5 (CH_2_), 59.8 (CH_2_), 108.3 (C), 110.7 (CH), 117.9 (CH), 119.3 (CH), 121.3 (CH), 126.8 (C), 127.2 (C), 127.5 (CH), 127.7 (CH), 129.6 (CH), 131.7 (C), 136.1 (C); HRESI-MS: calcd for C_15_H_18_N_2_ [M+H]^+^: 227.1543, found: 227.1543.

2-Benzyl-2,3,4,9-tetrahydro-1*H*-pyrido[3,4-*b*]indole (**3i**). The title compound was synthesized from (**2a**) (50 mg, 0.29 mmol) and benzyl chloride (0.05 mL, 0.44 mmol) to afford (**3i**) (22 mg, 29% yield) as a yellow oil; ^1^H-NMR (300 MHz, CDCl_3_) δ 2.83 (t, *J* = 5.2 Hz, 2H), 2.93 (t, *J* = 5.2 Hz, 2H), 3.66 (s, 2H), 3.78 (s, 2H), 7.07 (dt, *J* = 1.3, 7.1 Hz, 1H), 7.12 (dt, *J* = 1.3, 7.1 Hz, 1H), 7.27–7.44 (m, 6H), 7.47 (d, *J* = 6.0 Hz, 1H), 7.66 (br s, NH); ^13^C-NMR (75 MHz, CDCl_3_) δ 21.1 (CH_2_), 50.2 (CH_2_), 50.9 (CH_2_), 61.9 (CH_2_), 108.4 (C), 110.7 (CH), 117.9 (CH), 119.4 (CH), 121.4 (CH), 127.3 (CH), 128.4 (2CH), 129.2 (2CH), 131.7 (C), 136.0 (C), 138.2 (C).

#### 3.1.4. General Procedure for the Preparation of 2-substituted-1-phenyl-2,3,4,9-tetrahydro-1H-pyrido[3,4-b]indole (**3j**–**3k**)

1-Phenyl-2,3,4,9-tetrahydro-1*H*-pyrido[3,4-b]indole (**2b**) (1 eq) was dissolved in CH_2_Cl_2_ (1 mL/mmol), and NEt_3_ (3 eq) was added to the solution. Then, an acid halide or acyl halide was slowly added while stirring at room temperature for 24 h. Next, the product was extracted with CH_2_Cl_2_, and the organic layer was dried over anh. Na_2_SO_4_ and evaporated under reduced pressure. The crude product was purified using column chromatography (silica gel, 5% MeOH:CH_2_Cl_2_).

1-(1-Phenyl-3,4-dihydro-1*H*-pyrido[3,4-b]indol-2(9*H*)-yl)ethanone (**3j**). The title compound was synthesized from (**2b**) (117 mg, 0.47 mmol) and acetyl bromide (0.10 mL, 1.4 mmol) to afford (**3j**) (73 mg, 53% yield) as a brown solid; m.p. 236.6–237.8 °C (lit. 266 °C) [55]; ^1^H-NMR (300 MHz, CDCl_3_) δ 2.19 (s, 3H), 2.80–3.05 (m, 2H), 3.40–3.60 (m, 1H), 3.89 (dd, *J =* 2.6, 12.9 Hz, 1H), 7.02 (s, 1H), 7.13 (dt, *J =* 1.1, 7.1 Hz, 1H), 7.20 (dt, *J* = 1.1, 7.1 Hz, 1H), 7.28–7.40 (m, 5H), 7.54 (d, *J* = 7.1 Hz, 2H), 7.94 (br s, NH); ^13^C-NMR (100 MHz, CDCl_3_) δ 21.8 (CH_2_), 22.1 (CH_3_), 40.6 (CH_2_), 51.6 (CH), 109.9 (C), 111.1 (CH), 118.1 (CH), 119.6 (CH), 122.2 (CH), 126.6 (CH), 128.1 (C), 128.5 (2CH), 128.8 (2CH), 131.7 (C), 136.2 (C), 139.9 (C), 169.1 (C); HRESI-MS: calcd for C_19_H_18_N_2_O [M+H]^+^: 291.1492, found: 291.1499.

2-Octyl-1-phenyl-2,3,4,9-tetrahydro-1*H*-pyrido[3,4-*b*]indole (**3k**). The title compound was synthesized from (**2b**) (87 mg, 0.35 mmol) and 1-bromooctane (0.19 mL, 1.1 mmol) to afford (**3k**) (126 mg, 93% yield) as a brown solid; m.p. 88.9–90.5 °C; ^1^H-NMR (300 MHz, CDCl_3_) δ 0.86 (m, 3H), 1.10–1.34 (m, 11H), 1.53 (t, *J =* 6.8 Hz, 1H), 2.30–2.42 (m, 1H), 2.54–2.66 (m, 1H), 2.73 (dq, *J =* 4.3, 9.2 Hz, 1H), 2.89 (dt, *J =* 1.2, 3.9 Hz, 1H), 2.92–3.05 (m, 1H), 3.27–3.36 (m, 1H), 4.63 (s, 1H), 7.00–7.11 (m, 2H), 7.12–7.17 (m, 1H), 7.28–7.43 (m, 5H), 7.46–7.55 (m, 1H); ^13^C-NMR (100 MHz, CDCl_3_) δ 14.2 (CH_3_), 21.1 (CH_2_), 26.8 (CH_2_), 28.8 (CH_2_), 29.1 (CH_2_), 29.3 (CH_2_), 29.4 (CH_2_), 31.8 (CH_2_), 46.3 (CH_2_), 53.9 (CH_2_), 62.7 (CH), 108.9 (C), 110.6 (CH), 118.5 (CH), 119.4 (CH), 121.6 (CH), 127.1 (C), 128.1 (CH), 128.6 (2CH), 129.3 (2CH), 134.6 (C), 136.3 (C), 140.9 (C); HRESI-MS: calcd for C_25_H_32_N_2_ [M+H]^+^: 361.2638, found: 361.2640.

### 3.2. Antifungal Activity Assay

The in vitro antifungal activity of all tetrahydro-ß-carboline derivatives was evaluated on the phytopathogenic fungi of rice (*Oryza sativa*) against four fungi: *Bipolaris oryzae*, *Curvularia lunata*, *Fusarium semitectum,* and *Fusarium fujikuroi*. The tested fungi were cultured on potato dextrose agar (PDA) and incubated at 25 ± 2 °C for 7 days [56].

#### 3.2.1. Agar Well Diffusion Method

The well diffusion test was performed using Sabouraud dextrose agar. The solution of the synthesized compounds was dissolved in methanol and acetone for (**3b**, **3c**) to a final concentration of 20 mg/mL. The solvent of the tetrahydro-ß-carboline derivatives was used as the negative control, while Amphotericin B served as the positive control. The Sabouraud dextrose agar was poured into a Petri dish (9 cm in diameter) and allowed to cool. Each agar was cut out to 4 mm in diameter, and 50 µL of the test sample was placed in each well. The fungi were incubated at 30 ± 2 °C for 7 days. After the incubation period, the diameters of the fungal colonies were measured. All compounds and controls were performed in two duplicates. The mean and standard deviation were calculated.

#### 3.2.2. Minimum Inhibitory Concentration Test

MIC assays were determined using the broth microdilution method of the National Committee for Clinical Laboratory Standards (NCCLS 2002) [57]. Primarily, the phytopathogenic fungi were grown on PDA at 28 °C for 7 days to produce conidia. The fungal colonies were covered with 5 mL of normal saline, and the suspensions were made using a fine brush. The conidia were counted in a hematocytometer, which ranged from 2 × 10^6^ to 4 × 10^6^ spore/mL. Stock solutions of 20 mg/mL were dissolved in methanol and acetone for (**3b**, **3c**). Serial dilutions of the stock solutions were performed using Sabouraud dextrose broth glucose phenol red, with the final concentration range from 2.000 to 0.001953 mg/mL. A 100 µL aliquot of the stock solutions was transferred to each of the first wells, then 100 µL/well of each suspension was dispensed into a 96-well plate. The microdilution plates were incubated at 30 ± 2 °C for 7 days of incubation. All tests were performed in three replicates. The MICs were read and interpreted, and the endpoint determination reading was performed visually based on a comparison of the growth in the wells containing the stock solutions with that of the growth control. Amphotericin B was used as the standard control, while methanol and acetone were used as the negative control.

#### 3.2.3. Test for Inhibitory Activity against Fungal Radial Growth

Four concentrations of tetrahydro-ß-carboline (**3g**), 0, 100, 250, 500, and 1000 µg/mL, were tested for antifungal activity on the radial growth of the fungi on a PDA medium. The stock solutions were put in the center of a Petri dish and then added with 20 mL of melted PDA medium. The volume of the compound was added to a Petri dish and was adjusted according to the concentration tested. The Petri dishes were slightly shaken for a smooth distribution of the tested compound into the PDA medium. Amphotericin B (250 µg/mL) was used as the standard control. Additionally, plates without the compound were used as the negative control. A mycelia plug (5 mm diam.) of the fungi was taken from the edge of a 7-day old culture and placed in the center of the PDA. The cultures were incubated at 25 ± 2 °C for 7 days. The diameter of the fungal colony was measured and recorded on day 8 of inoculation and incubation. The inhibitory activity to the radial growth was determined according to the following formula:(1)$$\mathrm{IR}\left(\%\right)=\frac{\mathrm{DC}-\mathrm{DT}}{\mathrm{DC}}\times 100$$
IR = inhibitory activity against radial growth in percent.DC = diameter of fungal colony without compounds (control).DT = diameter of fungal colony treated with compound.

## 4. Conclusions

In summary, a series of tetrahydro-ß-carbolines (**2a**–**2h**) was synthesized from tryptamine (**1a**) or tryptophan methyl ester (**1b**) and various aldehydes for 1–2 h via Pictet-Spengler condensation (9–80% yields), followed by *N*-alkylation or *N*-acylation with alkyl halides or acyl halides, respectively, to afford 1,2-substituted THßCs (**3a**–**3k**) (13–93% yields). The synthesized compounds were tested against the plant pathogenic fungi *Bipolaris oryzae*, *Curvularia lunata*, *Fusarium semitectum*, and *Fusarium fujikuroi*, and the results showed that the substituent at the N2 position is essential for antifungal activity. The alkyl substituent at N2 increased the inhibitory activity, especially that of the *N*-octyl derivative (**3g**), which showed the highest antifungal activity with an MIC value range of 28–200 μg/mL. Additionally, compound (**3g**) displayed good inhibitory activity to the radial growth of tested fungi at a concentration of 100 μg/mL. Accordingly, this active compound (**3g**) showed broad range of antifungal activities *against* four plant pathogenic fungi in this study and *N*-alkyl THßCs could provide useful information for further development of novel THßCs fungicides.

## Data Availability

The data presented in this study are available on request from the corresponding author.

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
