# Peer review of "Synthesis and Investigation of Tetrahydro-β-carboline Derivatives as Inhibitors of Plant Pathogenic Fungi"

_molecules, 2021, doi:10.3390/molecules26010207_

Round 1

Reviewer 1 Report

GENERAL COMMENTS

The manuscript “Synthesis and Investigation of Tetrahydro-ß-carboline Derivatives as Inhibitors of Plant Pathogenic Fungi" shows interesting results and confirms that the substituent at the piperidine nitrogens plays an important role for increasing antifungal activities. I think that many researchers in this field may be interested to take a look on. However, the manuscript lacks of discussion. This section should be improved.

Minor comment

-Figures 1 and 2 should be fixed in only one image with part A and B.

Author Response

The discussion was added in Text with red and the figures 1 and 2 were combined as the figure 1

Reviewer 2 Report

The manuscript by Buaban, Phutdhawong et al.  describes the synthesis of a family of tetrahydro-β-carboline derivatives and their biological evaluations as inhibitors of plant pathogenic fungi. The synthetic procedures are based on the known Pictet-Spengler condensation, and some of the prepared compounds showed important antifungi activity, being the N-octyl derivative 3g the most active one. I believe that this paper would be acceptable for publication as an article in Molecules only after major revisions:

  1. The large differences in yields shown in Scheme 1 cannot be justified based on steric and electronic factors in most cases. Compare, for example, the yields for 3a versus 3b, 3f versus 3g, or 3j versus 3k. Given that derivative 3g, which happens to be the most active, is obtained with a low 15% yield, the yields for the synthesis and isolation of the desired products must be optimized.
  1.  Table 1 shows the results of the studies of in vitro zones of inhibition and MIC of the THbC derivatives on phytopathogenic fungi. However, this table of results is incomplete, with a high number of results classified as NA or NT. The authors must complete these studies and the results obtained must be included.
  2.  Given that all the synthesized compounds have a stereogenic center in their structures, the authors should consider the need to prepare and test at least one of them, the most active compound 3g, in enantiopure form and determining the activity of each enantiomer separately.
  3. In the case of 2h derivatives, their synthesis starting from the racemic mixture of tryptophan methyl ester, yields the desired compound as a mixture of 4 stereoisomers, which can have very different activities. Considering that L- and D-tryptophan methyl ester are commercially available in enantiopure form, they can be used as starting materials instead of the racemic sample, so that the synthesis and separation of each stereoisomer of 2h could be accomplished in order to determine the influence of chirality on antifungal activity.

Minors:
-  line 45 and line 61 in Figure 1: "2,2,2-TFA" must be   replaced by “TFA”.

-  line 63 in Figure 2:  "b-carboline" must be replaced by   “β-carboline”.

-  In the experimental section: "MeOD" must be replaced   by "CD3OD”.

- Throughout the entire experimental part, the authors  must modify the way of expressing the quantities of reactants and products, using a maximum of 2 decimal places. For this, it is suggested to use milligram instead of gram as the unit of mass.

- The Reference section must be revised and corrected. For example, references 8 and 13 are identical and in references 15 the number of the last page of the paper must be added.

Author Response

  1. The large differences in yields shown in Scheme 1 cannot be justified based on steric and electronic factors in most cases. Compare, for example, the yields for 3a versus 3b3f versus 3g, or 3j versus 3k. Given that derivative 3g, which happens to be the most active, is obtained with a low 15% yield, the yields for the synthesis and isolation of the desired products must be optimized.   Answer The yield of 3g, the most active product was optimized and yields of some of the compounds were optimized (as indicated as Red in Scheme1). Discussion according to low yield was added in Text (page 3 line 76-77 and 81-82).
  2. Table 1 shows the results of the studies of in vitro zones of inhibition and MIC of the THbC derivatives on phytopathogenic fungi. However, this table of results is incomplete, with a high number of results classified as NA or NT. The authors must complete these studies and the results obtained must be included. Answer According to Table 1, compounds with no zone of inhibition were not tested for MIC and N-substituted THbCs were tested for MIC because they showed zone of inhibitions for all tested phytopathogenic fungi but the C1 substituted THbCs were active against 2 pathogenic fungi. (Discussion was added in page 4 line 102-103).
  3. Given that all the synthesized compounds have a stereogenic center in their structures, the authors should consider the need to prepare and test at least one of them, the most active compound 3g, in enantiopure form and determining the activity of each enantiomer separately. Answer We have planed to prepare stereoselective C1 THbCs and tested for their activity. However, the N-substituted THbCs in this paper (as I understand) have no enantiomer since there is no chiral center.
  4. In the case of 2h derivatives, their synthesis starting from the racemic mixture of tryptophan methyl ester, yields the desired compound as a mixture of 4 stereoisomers, which can have very different activities. Considering that L- and D-tryptophan methyl ester are commercially available in enantiopure form, they can be used as starting materials instead of the racemic sample, so that the synthesis and separation of each stereoisomer of 2h could be accomplished in order to determine the influence of chirality on antifungal activity. Answer we used L-tryptophan methyl ester as starting material, thus 2 stereoisomers were obtained but we have planed to study the antifungal activity of each stereoisomers as discuss in page 4 line 109-110.
  5. All the minor errors have been revised and mark as red in text. 

Reviewer 3 Report

The article describes the synthesis and testing of a number of carboline derivatives as inhibitors of plant fungi. The synthesis was a simple well known 2 step process and in many cases proceeeded with low to moderate yields. It was determined that substitution at on the nitrogen was important for inhibitory effect. No motive (or mechanistic rational) was given for the design of these compounds other than compounds that contain this ring system are known to have a wide variety of biological effects.

I thought the paper was well written and easy to follow and the work seems well executed (experimentally). My only doubt is of the final conclusion. One compound (3g N-octyl) had 100% inhibitory effect as was seen as a lead compound for future development. However, the 100% inhibition came at very high concentrations (compare with amphotericin in table 2) and I would postulate adding any compounds at high concentrations will likely intefere and inhibit bacteria. The problem is that such high concentrations likely could not be used in real world situations as they are likely toxic. It would therefore have to be shown that such high concentrations could be used with a plant cell culture to remove this doubt.

Given these doubts I cannot agree the questions are warranted. Whilst as stated the work seems well executed, the conceptual rational behind (and conclusions drawn from the study) are not.

I would consider a revised version with the additional testing of the toxicity of the lead compound for example at various concentrations. Failing that I would not accept this manuscript in its present form. 

Author Response

One compound (3g N-octyl) had 100% inhibitory effect as was seen as a lead compound for future development. However, the 100% inhibition came at very high concentrations (compare with amphotericin in table 2) and I would postulate adding any compounds at high concentrations will likely intefere and inhibit bacteria. The problem is that such high concentrations likely could not be used in real world situations as they are likely toxic. It would therefore have to be shown that such high concentrations could be used with a plant cell culture to remove this doubt.

I could not agree more with your comments. Thus, the discussion was focused on the concentration of 100 µg/mL with can inhibit the growth of fungi at 37.4-68% with the exception of F. semitectum. Although, this compound (3g) was less active than the antifungal drug, it may give us the information for further development of antifungal agents in the future. (Page 4, line 111-118).

Round 2

Reviewer 1 Report

The paper can be accepted.

Reviewer 2 Report

I consider that with the modifications and corrections made by the authors, the work meets the necessary conditions to be published.

However, in those derivatives with a chiral center, the authors should consider the importance of chirality on activity. Taking into account that D-tryptophan is commercially available, the corresponding epimeric derivatives to L-tryptophan should be prepared in future works.

Reviewer 3 Report

The authors have improved somewhat the original version of the manuscript and therefore I would be willing to accept this revised version for publication.